# Canine Mammary Neoplasia Induces Variations in the Peripheral Blood Levels of CD20, CD45RA, and CD99

**DOI:** 10.3390/ijms24119222

**Published:** 2023-05-25

**Authors:** Makchit Galadima, Iuliia Kotova, Ronny Schmidt, Josep Pastor, Christoph Schröder, Joan Enric Rodríguez-Gil, Maria Montserrat Rivera del Alamo

**Affiliations:** 1Department of Animal Medicine and Surgery, Faculty of Veterinary Medicine, Universitat Autònoma de Barcelona, 08193 Bellaterra, Spain; makchitgaladima@gmail.com (M.G.); josep.pastor@uab.cat (J.P.); juanenrique.rodriguez@uab.cat (J.E.R.-G.); 2Sciomics GmbH, Karl-Landsteines-Straβe 6, 69151 Neckargemünd, Germany; iuliia90kotova@gmail.com (I.K.); ronny.schmidt@sciomics.de (R.S.); schroeder@sciomics.de (C.S.)

**Keywords:** canine mammary tumours, serum biomarkers, CD99, CD45RA, CD20

## Abstract

The idea of using tumour biomarkers as diagnostic tools is progressively increasing. Of these, serum biomarkers are of particular interest, as they can provide rapid results. In the present study, serum samples from 26 bitches diagnosed with mammary tumours, plus 4 healthy bitches, were obtained. The samples were analysed using CD antibody microarrays targeting 90 CD surface markers and 56 cytokines/chemokines. A total of five CD proteins, namely CD20, CD45RA, CD53, CD59, and CD99, were selected and further analysed, utilizing immunoblotting techniques to validate the microarray results. CD45RA showed a significantly lower abundance in the serum samples from the bitches carrying mammary neoplasia in comparison to the healthy animals. Regarding CD99, the serum samples from the neoplastic bitches showed it in a significantly higher abundance than those from the healthy patients. Finally, CD20 showed a significantly higher abundance in bitches carrying a malignant mammary tumour in comparison to healthy patients, but no differential expression between malignant and benign tumours was observed. According to these results, both CD99 and CD45RA are indicators of mammary tumour presence, but without distinguishing between malignant and benign.

## 1. Introduction

Mammary tumour is a life-threatening condition in both humans and bitches [1,2,3], although dogs display the highest incidence of mammary tumours among all the mammalian species [1,4,5]. Canine mammary tumour (CMT) is the second most common tumour after skin cancer in dogs [1,2,3,6], especially in intact bitches [6,7,8,9,10]. About 50% of CMTs are diagnosed as malignant and can metastasize to other organs through the vascular or lymphatics systems [8,11,12]. CMTs are prevalent globally, with a reported incidence rate of 200/100,000 dogs/year, showing lower incidences in countries that practice early spaying [8].

Mammary tumour development, like other neoplasia, is associated with inflammation [13,14], which is a normal quick response to acute tissue damage resulting from physical injury, ischemic injury, toxins, or other types of injury [13,14]. Whatever its origin, inflammation in the tumour microenvironment (TME), mainly chronic, has many cancer-promoting effects on and aids in the proliferation and survival of malignant cells, as well as promotes angiogenesis and metastasis [15]. The TME is composed of innate immune cells (including macrophages, neutrophils, mast cells, myeloid-derived suppressor cells, dendritic cells, and natural killer cells), adaptive immune cells (T and B lymphocytes), tumour cells, and their surrounding stroma, which consist of fibroblasts, endothelial cells, pericytes, and mesenchymal cells [13,14,15]. Immune cells provide diverse molecules to this TME, such as growth factors, proangiogenic factors, and extracellular-matrix-modifying enzymes, among others [16,17,18]. Thus, the link between inflammation, mainly chronic, and immune cells in tumorigenesis is evident.

To date, canine mammary diagnoses are usually performed by means of histopathology techniques. However, using tumour biomarkers as diagnostic tools has gained interest [11]. The promising mammary tumour biomarkers in both humans and canines are cancer-associated stroma (CAS) or TMEs, circulating tumour cells and tumour DNA (ctDNA), exosomes and miRNAs, and metabolites [19]. Variations in the expression of these markers may be helpful for achieving an early diagnosis, providing a prognosis, following the course of anti-tumoral therapy, predicting the response or resistance to specific therapies and the surveillant after primary surgery [11,20].

Nevertheless, CMT biomarkers are poorly studied and choosing the most appropriate biomarker remains the biggest challenge in optimizing tumour diagnoses [11,21]. Since early detection is crucial for the evolution of patients with mammary tumours, the determination of these biomarkers is key to evaluating the disease progression and response to treatments [22]. However, tissue biomarkers, although useful, do not provide an improvement in CMT in terms of immediacy, whereas serum biomarkers open up an interesting possibility of improving the celerity of the diagnosis. The literature focused on biomarkers in the peripheral blood is abundant in human research, but it is nearly absent when focusing on CMT.

Against this background, this study aimed at two different goals. The first objective aimed to determine the putative variations in the expressions of both the inflammatory and immune molecules in peripheral blood. On the other hand, it aimed to determine the potential of these molecules as serum biomarkers in canine mammary tumours. For these purposes, serum samples from 30 bitches were analysed through CD antibody microarrays targeting 90 CD surface markers and 56 cytokines/chemokines (Sciomics GmnH, Neckargemünd, Germany). Furthermore, the CD molecules showing differential expressions were validated using immunoblotting techniques.

## 2. Results

### 2.1. Animals and Histopathology

Twenty-six patients carrying mammary neoplasia were submitted to the teaching veterinary hospital at the Universitat Autònoma de Barcelona (Bellaterra, Spain) for surgical removal. After performing the surgery, tissue samples were submitted to the laboratory for a histopathology evaluation. The histology analyses yielded a total of 12 bitches with benign mammary tumours, 10 carrying mammary carcinoma, and 4 carrying both mammary carcinoma and adenoma. On the other hand, the 4 healthy bitches were proven, using histopathology, to have normal mammary tissue.

### 2.2. Microarrays

When comparing the samples from the healthy animals and the samples from the bitches carrying benign mammary tumours, 20 molecules recorded differential protein abundances in terms of either their log FC or *p* values (Table 1). However, only two proteins showed differential abundances in terms of both the log FC and *p*-value simultaneously, namely ITAX and CD177.

The proteins with positive log FC values showed a higher abundance in the samples from the bitches with benign mammary tumours, whereas those with negative values showed a higher abundance in the samples from the healthy bitches. |log FC| > 1 and > −1 indicated a differential abundance between the analysed groups. *p* < 0.05 indicated statistically significant differences.

When comparing the samples from the healthy animals and samples from the bitches carrying malignant mammary tumours, 36 molecules recorded differential protein abundances in terms of either their log FC or *p* values (Table 2). When considering both the log FC and *p* values, six proteins showed differential abundances in terms of both the log FC and *p* value simultaneously, namely CD20, CD53, ITAX, FCG3A, CD177, and MPRI.

The proteins with positive log FC values showed a higher abundance in the samples from the bitches with malignant mammary tumours, whereas those with negative values showed a higher abundance in the samples from the healthy bitches. |log FC| > 1 and >−1 indicated differential abundances between the analysed groups. *p* < 0.05 indicated statistically significant differences.

Finally, when comparing the samples from the bitches carrying malignant mammary tumours or benign mammary tumours, 17 molecules recorded differential protein abundances in terms of either their log FC or *p* values (Table 3). When considering both the log FC and *p* values, seven proteins showed differential abundances in terms of both the log FC and *p* value simultaneously, namely PDCD1, IFNA1, CD44, CD20, CD15, IL37, and TNR8.

The proteins with positive log FC values showed a higher abundance in the samples from the bitches with malignant mammary tumours, whereas those with negative values showed a higher abundance in the samples from patients carrying benign mammary tumours. |log FC| > 1 and >−1 indicated differential abundances between the analysed groups. *p* < 0.05 indicated statistically significant differences.

### 2.3. Immunoblotting

The microarrays analysis yielded an elevated number of differential proteins between the groups of both the CD surface markers and the cytokines/chemokines. Of these, the five most relevant CD molecules were selected for further analyses, utilizing Western blotting for validation, namely CD20, CD45RA, CD53, CD59, and CD99 (Figure 1).

The data obtained from the Western blotting were analysed using two different approaches. On the one hand, the data were analysed by distributing the samples in the healthy and neoplasia groups. On the other hand, the data were distributed in the healthy, benign mammary neoplasia, and malignant mammary neoplasia groups.

When comparing the healthy and neoplastic samples, CD99 showed a significant overexpression in the neoplastic samples (Figure 2D), whereas CD45RA showed significantly lower values in the serum from the bitches with mammary neoplasia (Figure 2D).

When comparing the healthy, benign, and malignant samples, CD99 showed a significant overexpression in the neoplastic samples, regardless of if they were benign or malignant, in comparison to the healthy samples (Figure 3D). In contrast, only the carcinoma samples showed a significant overexpression of CD20 (Figure 3E) in comparison to the healthy bitches. However, no significant difference between the benign and malignant samples for CD20 expression was observed.

## 3. Discussion

The present study demonstrated that the presence of mammary neoplasia in bitches induces modifications in the abundance of the cluster of differentiation (CD) 45RA, CD99, and CD20 in the peripheral serum. CD45RA and CD99 were under- and overexpressed in the peripheral serum from the bitches carrying CMTs, respectively, in comparison to the healthy bitches. When comparing benign mammary neoplasia, malignant mammary neoplasia, and healthy animals, CD20 showed a significant increase in the serum from the bitches with malignant CMTs in comparison to the healthy patients, but no differences were observed between malignant and benign CMTs.

CDs are cell surface molecules that provide targets for immunophenotyping [23] and are expressed in leukocytes. CD molecules are involved in diverse biological processes such as cell-to-cell communication and the stimulation of the immune response in front of foreign agents [24]. Focusing on the field of neoplasia, CD molecules have been described to be expressed in neoplasia tissue and have thus been proposed as good diagnosis and prognosis markers in both tissue and peripheral blood samples of neoplasia (see [24] for a review).

Specifically centring on the mammary gland, the presence of CD molecules has been described in both the healthy and neoplastic mammary tissues of women. In this sense, the ductal cellular layer of the mammary gland is composed of a relevant population of immune cells, which include CD8^+^ and CD4^+^ T cells [25,26]. Regarding mammary neoplasia tissue, several studies have evaluated the expression of CD molecules in human breast cancer (see [24,27] for reviews), but the literature on CMT is much less abundant [28,29,30].

In this respect, CD45 is a cell surface glycoprotein family, with CD45RA being one of its numerous isoforms [31,32,33]. Monocytes and macrophages labelled with this molecule are known to infiltrate tumour microenvironments [34,35]. Mammary tumours being infiltrated with CD45 have been previously described, although scarcely, in mice, also showing an increased tissue infiltration [36]. Thus, there is a clear correlation between CD45 and tumorigenesis. As stated above, CD45RA was under-expressed in the peripheral blood of the bitches carrying CMTs. A feasible hypothesis for this could be the fact that CD45^+^ cells were displaced to the mammary tumour, maybe in response to the inflammatory reaction occurring in the tumoral area. In fact, the migration of tumorigenic molecules from the peripheral blood to tumorigenic tissue has been already described in the literature on ovarian neoplasia [37]. Therefore, it seems logical to think that CD45^+^ labelled mono-macrophages may migrate from the peripheral blood to tumour tissue. This hypothesis would be reinforced by the fact that CD45^+^ cells have been related to tumoral vasculogenesis [38]. However, further research is needed to confirm this hypothesis, since CD45 expression has not yet been evaluated in CMT.

CD99 is a surface glycoprotein which is expressed in endothelial and haematopoietic lineage cells, among others [39]. In inflammatory processes, this molecule regulates the adhesion and transendothelial migration of haematopoietic cells [40,41]. Its expression has been demonstrated in several types of neoplasia, including mammary tumours in women [42], being suggestive of an increased invasiveness [43], tumour migration [44], and malignancy [45]. Nevertheless, the role of CD99 in tumour development, growth, migration, and metastasis is still under discussion, as it is considered to be both an oncogenic and tumour suppressor, depending on the type of neoplasia (see [39] for a review). In this sense, under-regulation has been associated with tumour progression in osteosarcoma and gastric cancer [46], whereas the over-expression of CD99 has been associated with a higher migration, tumour growth, and metastasis in Ewing’s sarcoma [47,48], given that the present results are in agreement with the latter.

Specifically focusing on the peripheral blood, the overexpression of this molecule has been described in neoplasia such as Ewing’s sarcoma [49,50]. According to the literature, this is the first time that the peripheral blood levels of CD99 have been analysed for mammary tumours in any species. Remarkably, the overexpression of CD99 in the peripheral blood, together with the low levels of CD45 (CD99^+^CD45^−^), has been related to a very poor prognosis in children experiencing Ewing’s sarcoma [50]. However, this is not applicable in the present study, since our results suggest that CD99^+^CD45^−^ expression is associated with the presence of CMT, but do not provide a differential diagnosis between malignant and benign CMT. Thus, it cannot be related to tumour progression or metastasis.

CD99 is also related to inflammation (see above). During an inflammatory response, the affected tissue releases pro-inflammatory mediators, which, in turn, induce local changes in the endothelium that will be translated into leukocyte extravasation [51]. This leukocyte extravasation is facilitated by CD99, among other molecules [41]. The association between tumorigenesis and inflammation is well known, and mammary tumours are not an exception. Thus, this increased expression of CD99 in the serum samples from the bitches with CMTs may be related to tumorigenic inflammation.

CD20 is a surface protein expressed in B cells during their development [52]. It is thought to be involved in the control of cell growth by means of regulating calcium influx [53]. In situ, CD20 is overexpressed in ductal carcinoma in women [27] and a correlation between tissue and peripheral expression has also been found in breast cancer [54]. However, CD20 is poorly described in CMT, being described only in canine primary breast lymphoma as an unexpected finding [55], with no literature focusing on the peripheral levels of this molecule. The role of CD20 in breast cancer is controversial. While some studies have affirmed an association between CD20^+^ B-cells and a favourable prognosis in invasive breast cancer [56], others have observed an association with a poor prognosis [57] and reduced disease-free survival [58].

Circulating levels of CD20 have been associated with better prognoses in women with breast cancer [59]. In the present study, no such conclusion can be reached, since the values were higher in the bitches with malignant mammary tumours, but these values were significantly higher only in comparison to the healthy bitches. Benign mammary tumours yielded a lower mean value in comparison to malignant mammary tumours, but this difference was not significant, probably due to the high dispersion of the data in the latter group. When analysing the results individually, it can be observed that half of the samples yielded low values for the peripheral CD20 and half of the samples yielded considerably high values. We can only hypothesize to figure out the reason for this data dispersion. As stated above, CD20 has yielded controversial results in women’s breast cancer, which is probably associated with the different types of mammary neoplasia. It seems logical to think that this could be also a feasible explanation in CMT. No information other than the histology classification was considered. Thus, maybe considering other parameters, such as the presence/absence of metastasis, invasiveness of the tumour, and presence/absence of more than one tumour of the same/different type in the patient, among others, would yield a more accurate result.

Chronic inflammation, in tight cooperation with the immune system, has long been associated with cancer development, including, of course, breast cancer. Chronic inflammation enhances angiogenesis and tissue invasion, which will promote cellular proliferation and cancer progression [15]. On the other hand, it releases carcinogenic molecules [60] that recruit immune and regulatory cells [61]. Immune cells are components of the normal mammary tissue (see above). The appearance of a tumorigenic process provokes modifications in both the qualitative and quantitative compositions of these immune cells in the mammary tissue [62,63]. The present results reinforce the hypothesis that the immune system, in cooperation with inflammation, is involved in CMT.

Although much research has been performed to date, mammary neoplasia is still surrounded by many unanswered questions. Many endogenous factors, some of them associated with chronic inflammation and the immune system, are involved in the development, growth, and migration of these tumours in an intertwining way that is not completely elucidated at all. As more studies are performed, more questions arise, so further research is needed to improve the understanding of mammary tumours, and, in turn, improve the current diagnostic, prognostic, and treatment tools.

One of the objectives of the present study was to evaluate CD surface molecules and cytokines/chemokines as potential CMT biomarkers. Our results allow for the establishment of the presence of new players in this macro-game, but do not allow for the establishment of new diagnostic biomarkers, making further research mandatory.

## 4. Materials and Methods

### 4.1. Animals and Sampling

A total of 30 bitches were included in the study. Twenty-six of them were referred to the Fundació Hospital Clinic Veterinari at the Autonomous University of Barcelona (UAB, Bellaterra, Spain) for mammary tumour resections, while 4 bitches referred for elective ovariohysterectomies were used as a control group. The ages of the females with mammary tumours ranged from 7 to 14 years (mean age: 9.72 years), whereas those of the control bitches ranged from 3 to 5 years (mean age: 3.75 years). The samples were always obtained under the signed consent of the owner and followed the guidelines of the Ethical Committee Animal Care and Research, Autonomous University of Barcelona (Bellaterra, Cerdanyola del Vallès, Spain). The specimens were obtained following the guidelines of the Ethical Committee of Animal Care and Research of the UAB (protocol CEEAH number 1127, 20 March 2012).

Before surgery, 3–5 mL of blood was collected through a venipuncture of the jugular vein, deposited into a tube without anticoagulant, and allowed to clot at room temperature. Thereafter, it was centrifuged at 2000× *g* for 10 min and the serum was aspirated with a Pasteur pipette into Eppendorf tubes. The blood samples were stored at −80° until they were needed for the microarray and immunoblotting techniques.

After surgical resection, the mammary tumours were fixed with 10% paraformaldehyde (Sigma-Aldrich, Barcelona, Spain) and submitted to the laboratory for the histological diagnosis and typification of the mammary tumours. In the control group, a 1 cm × 1 cm specimen was obtained from the mammary gland of each bitch and further submitted for a histology examination to check for the absence of mammary tumours.

### 4.2. Microarrays Technique

First, the protein concentrations were determined using a Bicinchoninic Acid (BCA) assay with a BCA commercial kit (Thermo Fisher Scientific, Dreieich, Germany), following the supplier’s instructions. Briefly, the serum samples were labelled at an adjusted protein concentration for one hour with scioDye 1 and scioDye 2. Then, the reaction was stopped by adding hydroxylamine. The excess of dye was removed 30 min later and the buffer was exchanged for 1x phosphate-buffered saline (PBS).

Immediately, the samples were analysed in a dual-colour approach using a reference-based design on 30 scioCD antibody microarrays (Sciomics GmbH, Neckargemünd, Germany). These microarrays specifically targeted for 93 different CD surface markers and 25 cytokines/chemokines with 256 monoclonal antibodies (Table A1). Each antibody was represented on the array in four replicates. The arrays were blocked with scioBlock (Sciomics GmbH, Neckargemünd, Germany) on a Hybstation 4800 (Tecan, Austria) and the samples were then incubated competitively using a dual-colour approach. After incubation for three hours, the slides were thoroughly washed with 1x PBSTT, rinsed with 0.1x PBS, as well as with water, and subsequently dried with nitrogen.

Slide scanning was conducted using a Powerscanner (Tecan, Austria) with identical instrument laser power and adjusted PMT settings. Spot segmentation was performed with GenePix Pro 6.0 (Molecular Devices, Union City, CA, USA). Once the microarray technique was applied, the proteins yielding differentiation values among the different types of samples were validated by means of immunoblotting techniques.

### 4.3. Immunoblotting

First, the protein concentrations of the serum samples were determined through the Bradford method [64], using the commercial kit Bio-Rad Protein Dye Reagent (BioRad, Hercules, CA, USA). Afterwards, the samples were diluted at a final concentration of 1 mg/mL with a homogenization buffer (TRIS HCl 50 mM, EDTA 1 mM, EGTA 10 mM, DTT 25 mM, 1.50% Triton x100, PMSF 1 mM, leupeptin 10 μg/mL, orthovanadate 1 mM, and benzamidine 1 mM). A total of 10 mg of protein from each sample was loaded.

The immunoblotting protocol was performed according to Sirois and Dore [65]. Briefly, the proteins were separated by SDS-PAGE and, after the electrophoresis, transferred using a Trans-Blot Turbo Transfer System (BioRad, El Prat del Llobregat, Spain). The transferred membranes were then probed against the different primary antibodies. Then, the detection was performed by using the corresponding secondary antibody. Afterwards, the membranes were incubated with Western Blotting Luminol Reagent (Santa Cruz Biotechnology, INC, Heidelberg, Germany) for 5 min and further exposed to a radiograph film.

All the sera were assessed using immunoblotting and two replicates for each sample were performed. The densitometric analysis included the duplicated values obtained from each sample.

### 4.4. Primary Antibodies

Anti-human CD20, anti-human CD45RA, anti-human CD53, anti-human/-mouse CD59, and anti-human CD99 antibodies were used in this experiment as the primary antibodies. They were purchased from ImmunoTools (Friesoythe, Germany). The membranes were incubated overnight at 4 °C, with each corresponding antibody at a 1:1000 (*v*/*v*) dilution.

### 4.5. Secondary Antibodies

After exposure to the primary antibodies, the membranes were incubated with a polyclonal rabbit anti-mouse secondary antibody at a 1:500 (*v*/*v*) dilution. The secondary antibody was purchased from Dako (Glostrup, Denmark).

### 4.6. Image Analysis

The slide scanning of the microarrays was performed utilizing Powerscanner (Tecan, Austria) equipment with identical instrument laser power and adjusted PMT settings. The spot segmentation was performed with Gene Pix Pro 6.0 (Molecular Devices, Union City, CA, USA).

The quantification of the proteins on the X-ray film from the immunoblotting was performed through the ImageJ software (URL: https://imagej.en.softonic.com/?ex=DINS-635.2; accessed on 2 December 2022).

### 4.7. Statistical Analysis

The acquired raw data from the microarrays technique were analysed using the linear models for the microarray data (LIMMA) package of R-Bioconductor after uploading the median signal intensities. A specialised invariant Lowess method was applied for the normalisation. For the analysis of the samples, a one-factorial linear model was fitted with LIMMA, resulting in a two-sided *t*-test or F-test based on the moderated statistics.

All the presented *p* values were adjusted for multiple testing by controlling the false discovery rate according to Benjamini and Hochberg [66]. The proteins were defined as differential for a |log FC| > 1 and a *p* value < 0.05. The differences in the protein abundances between the different samples or sample groups were presented as log-fold changes (log FC) calculated for the basis 2. In a study comparing samples versus controls, a log FC = 1 means that the sample group has, on average, a 2^1^ = 2-fold higher signal than the control group. log FC = −1 stands for 2^−1^ = 1/2 of the signal in the sample, as compared to the control group.

The raw data obtained from the immunoblotting technique were analysed by applying a Shapiro–Wilk test, which was used to study the normality distribution. When a parameter did not follow a normal distribution, a non-parametric test was used to assess the differences between the groups. The Kruskal–Wallis test was used when more than two variables were compared with multiple comparison tests and the Mann–Whitney test when two variables were compared. The correlations between the parameters were studied using Pearson or Spearman correlation coefficients, according to their distribution. The statistical analysis was performed using GraphPad Prism version 8.0.0 for Windows, GraphPad Software, San Diego, CA, USA, (URL: www.graphpad.com).

## 5. Conclusions

Canine mammary tumours induce variations in the peripheral concentrations of CD20, CD45RA, and CD99. However, these differences are only significant between animals carrying mammary tumours and healthy bitches, thus not allowing for a differentiation between malignant and benign CMT. Further research is warranted to determine the novel serum biomarkers that allow for more accurate and celeritous diagnoses.

## Figures and Tables

**Figure 1 ijms-24-09222-f001:**
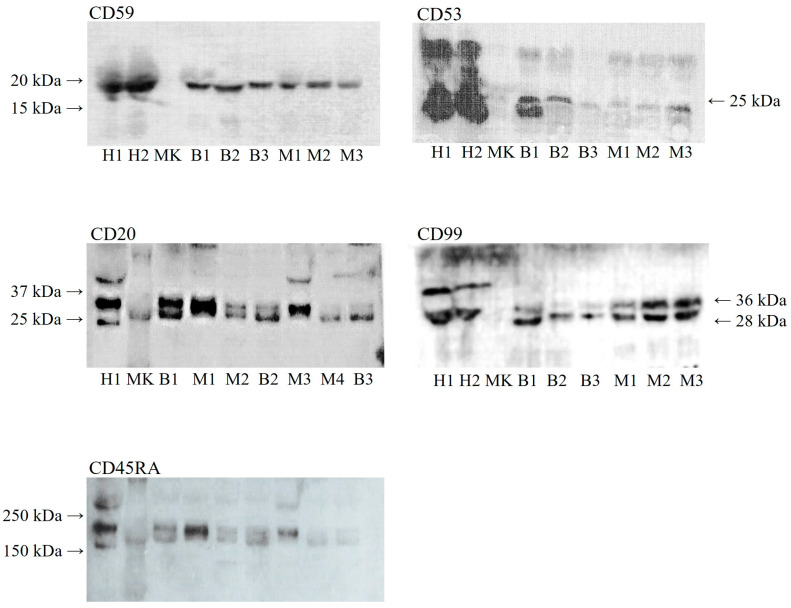
Western blotting images. H: healthy bitches; B: bitches carrying benign mammary tumours; M: bitches carrying malignant mammary tumours; and MK: molecular weight marker.

**Figure 2 ijms-24-09222-f002:**
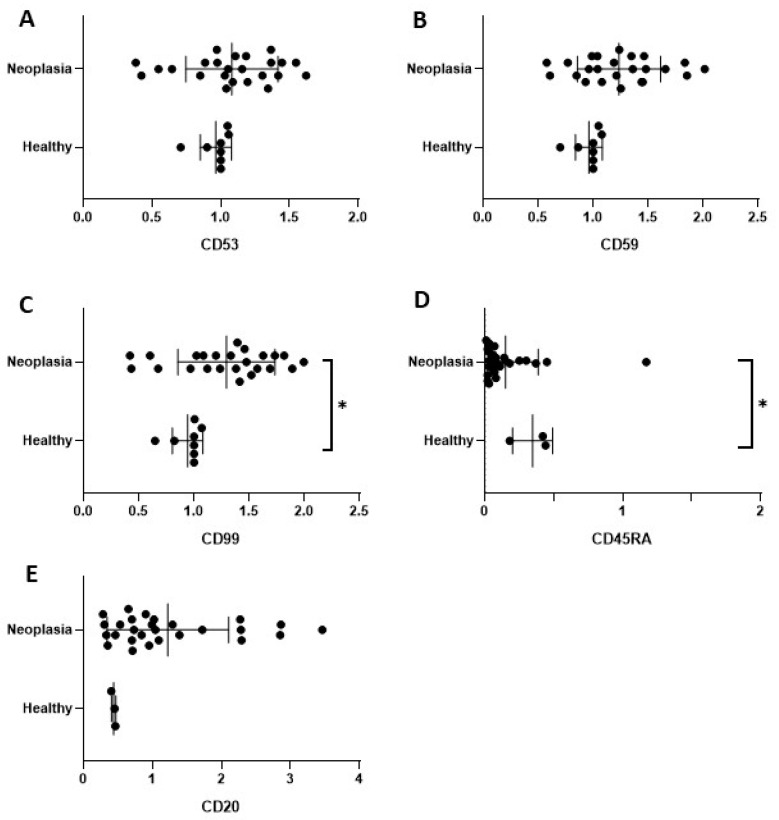
Comparative expression of peripheral blood levels for CD53 (**A**), CD59 (**B**), CD99 (**C**), CD45RA (**D**), and CD20 (**E**) between healthy (n = 4) and mammary neoplasia (n = 26) bitches. Results are expressed in terms of mean ± SD from two immunoblotting replicates. Statistically significant (*p* < 0.05) differences are marked with asterisks.

**Figure 3 ijms-24-09222-f003:**
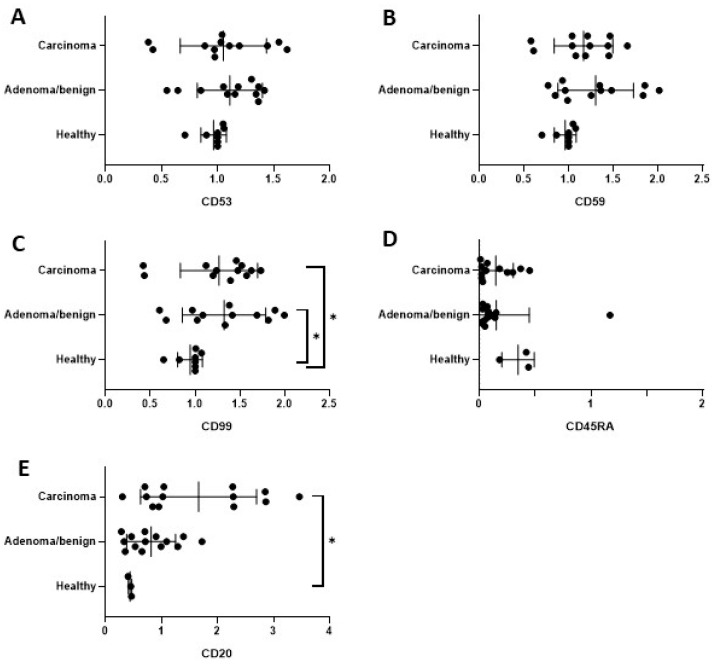
Comparative expression of peripheral blood levels for CD53 (**A**), CD59 (**B**), CD99 (**C**), CD45RA (**D**), and CD20 (**E**) among healthy (n = 4), benign (n = 12), and malignant mammary (n = 14) neoplasia bitches. Results are expressed in terms of mean ± SD from two immunoblotting replicates. Statistically significant (*p* < 0.05) differences are marked with asterisks.

**Table 1 ijms-24-09222-t001:** Proteins with differential abundance, benign tumours, cancer (n = 12) vs. healthy (n = 4) animal.

Protein	Antibody ID	log FC	AveExp	Adj. *p*-Val	Uniport-Link
CD20	S0288	1.51	13.45	0.270	P11836
ICAM1	S0333	1.25	11.50	0.920	P05362
CD53	S0332	1.23	13.91	0.650	P19397
ICAM1	S0334	1.22	12.28	0.920	P05352
DAF	S0335	1.08	9.49	0.370	P08174
MCP	S0325	1.05	12.60	0.960	P15529
CD20	S0286	−0.96	11.50	0.039	P11836
CD139	S0438	−1.01	14.31	0.950	
PDCD1	S0003	−1.01	12.57	0.960	Q15116
IL1β	S0387	−1.04	14.53	0.950	P01584
IL6	S0530	−1.06	10.01	0.490	P05231
CCL3	S0393	−1.07	14.54	0.920	P10147
IL16	S0529	−1.09	11.88	0.720	Q14005
CD86	S0356	−1.10	10.98	0.960	P42081
TGM2	S0405	−1.14	9.07	0.650	P21980
CD79A	S0354	−1.15	11.66	0.920	P11912
ITAX	S0268	−1.22	11.04	0.036	P20702
KIT	S0001	−1.24	13.51	0.960	P10721
CD24	S0294	−1.33	10.84	0.960	P25063
CD177	S0368	−1.34	10.62	0.039	Q8N6Q3
MPRI	S0369	−1.42	11.61	0.090	P11717
IFNA1	S0402	−2.10	13.94	0.720	P01562

**Table 2 ijms-24-09222-t002:** Proteins with differential abundance, malignant tumours, cancer (n = 14) vs. healthy animal (n = 4).

Protein	Antibody ID	log FC	AveExp	Adj. *p*-Val	Uniport-Link
CD20	S0288	2.57	13.45	0.002	P11836
MCP	S0325	2.05	12.60	0.820	P15529
CD3	S0407	2.05	12.86	0.290	P07766
PD1L1	S002	1.96	12.88	0.820	Q9NZQ7
CD53	S0332	1.92	13.91	0.040	P19397
AMPN	S0378	1.54	13.32	0.710	P15144
ICAM1	S0.334	1.47	12.28	0.480	P05362
CD44	S0424	1.43	14.10	0.190	P16070
CD22	S0292	1.40	11.35	0.420	P20273
CD99	S0360	1.31	11.75	0.430	P14209
CD8A	S0254	1.15	12.13	0.480	P01732
CD44	S0315	1.12	12.40	0.520	P16070
EGLN	S0363	1.02	11.72	0.820	P17813
CD20	S0286	−0.87	11.50	0.049	P11836
CEAM6	S0503	−1.00	12.45	0.630	P40199
IL7	S0388	−1.01	13.42	0.910	P13232
IL16	S0529	−1.02	11.88	0.470	Q14005
CEAM1	S050	−1.02	12.76	0.480	P13688
IL8	S0389	−1.08	11.41	0.190	P10145
LFA3	S0340	−1.10	12.29	0.910	P19256
CEAM5	S0348	−1.12	12.35	0.500	P06731
IL8	S0475	−1.15	10.33	0.420	P10145
ITAX	S0268	−1.16	11.04	0.019	020702
CD24	S0294	−1.16	10.84	0.820	P25063
NTF4	S0397	−1.22	13.88	0490	P34130
CCL7	S0394	−1.22	13.59	0.710	P80098
FCG3A	S0277	−1.23	10.50	0.011	P08637
CD177	S0368	−1.24	10.62	0.044	Q8N6Q3
CD53	S0331	−1.35	12.03	0.250	P19397
TGM2	S0405	−1.36	9.07	0.230	P21980
CCL11	S0382	−1.37	14.23	0.370	P51671
IL15	S0391	−1.39	14.48	0.290	P40933
CD139	S0438	−1.42	14.31	0.480	
IL1b	S0387	−1.43	14.53	0.480	P01584
BDNF	S0381	−1.50	14.15	0.400	P23560
IL18	S0392	−1.52	14.34	0.290	Q14116
MPRI	S069	−1.65	11.61	0.017	P11717
CCL3	S0393	−1.75	14.54	0.230	P10147
CD86	S0356	−2.12	10.98	0.390	P42081

**Table 3 ijms-24-09222-t003:** Proteins with differential abundance, malignant (n = 14) vs. benign tumours (n = 12).

Protein	Antibody ID	log FC	AveExp	Adj. *p*-Val	Uniport-Link
PDCD1	S0003	1.82	12.57	0.012	Q15116
IFNA1	S0402	1.72	13.94	0.008	P01562
CD44	S0315	1.31	12.40	0.007	P16070
KIT	S0001	1.24	13.51	0.370	P10721
CD3	S0407	1.11	12.86	0.170	P07766
PD1L1	S0002	1.08	12.88	0.550	Q9NZQ7
CD20	S0288	1.07	13.45	0.011	P11836
MCP	S0325	1.00	12.60	0.640	P15529
IL37	S0473	−0.42	10.27	0.012	Q9NHZ6
PTPRC	S0323	−068	11.29	0.012	P08575
CD38	S0308	−0.78	10.15	0.011	P28907
IL8	S0475	−0.80	10.33	0.021	P10145
NCAM1	S0337	−0.94	11.60	0.013	P13591
CD86	S0356	−1.03	10.98	0.200	P42081
CD15	S0273	−1.04	11.41	0.012	
LFA3	S0340	−1.04	12.29	0.550	P19256
IL37	S0474	−1.11	11.36	0.008	Q9NZH6
TNR8	S0302	−1.13	12.34	0.021	P28908

## Data Availability

Data are available at the authors’ request.

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
