# Peer review of "Canine Mammary Neoplasia Induces Variations in the Peripheral Blood Levels of CD20, CD45RA, and CD99"

_ijms, 2023, doi:10.3390/ijms24119222_

Round 1

Reviewer 1 Report

The article “Canine mammary neoplasia induces variations in peripheral blood levels of CD20, CD45RA and CD99” by Galadima et al., is a simple study which aims to introduce some CDs as potential biomarkers for rapid diagnosis of CMTs. The manuscript is poorly written, organized and presents a lot of fundamental flaws in experiment design. I have provided an exhaustive list of comments that the authors absolutely need to work upon to bring this manuscript into a presentable state. The article does not meet the standards of the journal in its current state. As such, I would have to reject this article at this stage and urge the authors to work on my concerns.

Major comments –

1.     I am really confused with the number of animals used for the experiments. In the abstract, the authors say 24 bitches with tumors and 4 healthy bitches (Total = 28). Then in the materials and methods they say 34 bitches were included in the study. In the results 2.1, they say 26 bitches with tumors and do not mention any number for healthy number. This should be rectified and more importantly, the exact number of animals or samples from animals used for experiments should be clearly mentioned. As an example, there is no information on how many serum samples were analyzed for the results shown in table 1 and 2. If the original number of animals was consistent, I would expect, 26 + 4. However, now I am not sure.

2.     The abstract can be made shorter. It also needs a rework due to poor use of English, especially in the first 4 sentences. As an example, the sentence “The interest in tumorigenesis biomarkers as diagnostic tools is progressively increasing, being serum biomarkers of special interest.” Can be re-written as “The idea of using tumor biomarkers as diagnostic tools is progressively increasing. Of these, serum biomarkers are of particular interest as they can provide rapid results.”

Further, the sentences that talk about CD99 and CD20 can be combined into 1 as they say the same thing.

3.     Introduction also suffers from poor use of English and needs to be cut down as well. It is too long. Please leave out the paragraph “Lately, a link between inflammation and the immune system has gained relevance as a pro- and an anti-tumorigenic factor [19-23]. Two different pathways of this inflammation-immunity interaction have been proposed. Thus, whereas anti-tumorigenic function of immunity is related with the immunological sculpting of the tumour and exerts an immunosurveillance role, pro-tumorigenic inflammation is known to promote neoplasia development by modulating the tumour microenvironment, blocking the anti-tumour immunity among other pro-tumorigenic functions [24].” Its contents are irrelevant.

4.     Please use a free tool like Grammarly to correct most of your English statements and typographical errors throughout the MS. Alternatively, please consult with a native English speaker for proper construction of sentences throughout the manuscript.

5.     There is no mention of the selection criteria for the 5 CDs looked at with immunoblotting. The results section with immunoblotting should start with a brief description of how the 5 CDs were agreed upon depending on the microarray data.

6.     Figure 1 is extremely poorly done. The H, B and M samples should be labelled with numerals like H1, H2, M1, M2 etc. so that the reader knows whether the same samples were used to look at different CDs. This would help maintain consistency between experiments and provide a better base for densitometry shown in figure 2. Please rework the figure.

7.     Please clearly mention how many western blots did you consider and whether all serum samples were used for densitometric analyses shown in figure 2 and 3. It is neither mentioned in the text, nor the figure legend. Doesn’t seem like the case when taking figure 1 into account, so please mention how many samples were used for densitometry.

Minor comments –

1. Please re-label tables 1, 2 and 3 as –

Table 1. Proteins with differential abundance, benign tumors, cancer vs healthy animal

Table 2. Proteins with differential abundance, malignant tumors, cancer vs healthy animal

Table 3. Proteins with differential abundance, benign vs malignant tumors

Please use a free tool like Grammarly to correct most of your English statements and typographical errors throughout the MS. Alternatively, please consult with a native English speaker for proper construction of sentences throughout the manuscript.

Author Response

The article “Canine mammary neoplasia induces variations in peripheral blood levels of CD20, CD45RA and CD99” by Galadima et al., is a simple study which aims to introduce some CDs as potential biomarkers for rapid diagnosis of CMTs. The manuscript is poorly written, organized and presents a lot of fundamental flaws in experiment design. I have provided an exhaustive list of comments that the authors absolutely need to work upon to bring this manuscript into a presentable state. The article does not meet the standards of the journal in its current state. As such, I would have to reject this article at this stage and urge the authors to work on my concerns.

Major comments –

  1. I am really confused with the number of animals used for the experiments. In the abstract, the authors say 24 bitches with tumors and 4 healthy bitches (Total = 28). Then in the materials and methods they say 34 bitches were included in the study. In the results 2.1, they say 26 bitches with tumors and do not mention any number for healthy number. This should be rectified and more importantly, the exact number of animals or samples from animals used for experiments should be clearly mentioned. As an example, there is no information on how many serum samples were analyzed for the results shown in table 1 and 2. If the original number of animals was consistent, I would expect, 26 + 4. However, now I am not sure.

Answer: The reviewer is completely right when stating that the number of females included in the study was confusing. These mistakes have been corrected in the body of the manuscript. The total number of bitches was 30 (26 with CMT + 4 healthy). In addition, the number of samples has been added to the legend of tables 1, 2 and 3. Finally, the number of healthy bitches in “Results 2.1” has been included in line 87.

  1. The abstract can be made shorter. It also needs a rework due to poor use of English, especially in the first 4 sentences. As an example, the sentence “The interest in tumorigenesis biomarkers as diagnostic tools is progressively increasing, being serum biomarkers of special interest.” Can be re-written as “The idea of using tumor biomarkers as diagnostic tools is progressively increasing. Of these, serum biomarkers are of particular interest as they can provide rapid results.”

Answer: The abstract has been slightly shortened and modified according to your suggestion.

Further, the sentences that talk about CD99 and CD20 can be combined into 1 as they say the same thing.

Answer: Unfortunately. I must disagree with this comment. CD99 showed higher expression in all neoplastic bitches when comparing with healthy bitches, whereas CD20 showed significant differences only between bitches carrying a malignant CMT and healthy bitches.

  1. Introduction also suffers from poor use of English and needs to be cut down as well. It is too long. Please leave out the paragraph “Lately, a link between inflammation and the immune system has gained relevance as a pro- and an anti-tumorigenic factor [19-23]. Two different pathways of this inflammation-immunity interaction have been proposed. Thus, whereas anti-tumorigenic function of immunity is related with the immunological sculpting of the tumour and exerts an immunosurveillance role, pro-tumorigenic inflammation is known to promote neoplasia development by modulating the tumour microenvironment, blocking the anti-tumour immunity among other pro-tumorigenic functions [24].” Its contents are irrelevant.

Answer: This paragraph has been removed following your advice

  1. Please use a free tool like Grammarly to correct most of your English statements and typographical errors throughout the MS. Alternatively, please consult with a native English speaker for proper construction of sentences throughout the manuscript.

Answer: The manuscript has been reviewed through the Grammarly tool following the reviewer’s advice.

  1. There is no mention of the selection criteria for the 5 CDs looked at with immunoblotting. The results section with immunoblotting should start with a brief description of how the 5 CDs were agreed upon depending on the microarray data.

Answer: Thanks for your comment. The sentence has been added to the body of the manuscript.

  1. Figure 1 is extremely poorly done. The H, B and M samples should be labelled with numerals like H1, H2, M1, M2 etc. so that the reader knows whether the same samples were used to look at different CDs. This would help maintain consistency between experiments and provide a better base for densitometry shown in figure 2. Please rework the figure.

Answer: The figure has been modified according to your suggestion.

  1. Please clearly mention how many western blots did you consider and whether all serum samples were used for densitometric analyses shown in figure 2 and 3. It is neither mentioned in the text, nor the figure legend. Doesn’t seem like the case when taking figure 1 into account, so please mention how many samples were used for densitometry.

Answer: All sera samples were evaluated by means of immunoblotting twice. This information was included in the body of the manuscript as well as the figures legend.

Minor comments –

  1. Please re-label tables 1, 2 and 3 as –

Table 1. Proteins with differential abundance, benign tumors, cancer vs healthy animal

Table 2. Proteins with differential abundance, malignant tumors, cancer vs healthy animal

Table 3. Proteins with differential abundance, benign vs malignant tumors

Answer: Tables have been re-labeled following your suggestion

Comments on the Quality of English Language

Please use a free tool like Grammarly to correct most of your English statements and typographical errors throughout the MS. Alternatively, please consult with a native English speaker for proper construction of sentences throughout the manuscript.

Answer: see the answer on bullet 4

Author Response

Galadima et al. bring to our attention a very original research on serum samples biomarkers for canine mammary tumors. A CD antibody microarray targeting 90 CD surface markers and 56 cytokines/chemokines was used for the analysis in serum samples from 30 bitches. They found 3 possible molecular serum targets: CD20, CD99 and CD45RA. Materials and methods are well explained and the results are described clearly and precisely. I think the work is suitable for publication in International Journal of Molecular Science with only some minor changes: 

Dear reviewer,

Thanks for your kind evaluation of our work. Here are our answers to your queries.

Introduction: it’s too long, please shorten it.

Answer: The introduction has been shortened following your advice

Introduction, page 2, line 59-61: this concept is wrong. In my view histopathology techniques are not time consuming and a diagnosis can be reached without removal of the entire mass through a fine needle aspiration or fine needle biopsy. Please modify accordingly. 

Answer: The sentence has been modified accordingly

After the Introduction section, you should place the material and methods section and not the results. Please respect the standard sequence of paragraphs for an original research: 

  1. Introduction
  2. Materials and Methods
  3. Results
  4. Discussion

Answer: Thanks for your comment. You are right when stating that the usual sequence of chapters is introduction-M&M-results- discussion. However, the draft plate offered by this journal follows this particular sequence.

Discussion: please do a figure comparing for CD45, CD99 and CD20 the main literature findings with the main results of the study.  

Answer: Dear reviewer, I must apologize but I’m not quite sure of what you are exactly requesting. Could it be so that you want us to include a table in the discussion chapter?

Round 2

Reviewer 1 Report

The authors have made a considerable effort to improve the manuscript as per the suggestions. I am ok with accepting the current version for publication.